# Hatching of whipworm eggs induced by bacterial contact is serine-protease dependent

David Goulding[1☉], Charlotte Tolley[2☉], Tapoka T. Mkandawire[1¤a], Stephen R. Doyle[1], Emily Hart[2], Paul M. Airs[2], Richard K. Grencis[3], Matthew Berriman[1¤b]*, María A. Duque-Correa[1¤c]*

**1** Wellcome Sanger Institute, Wellcome Genome Campus, Hinxton, United Kingdom, **2** Cambridge Stem Cell Institute, University of Cambridge, Cambridge, United Kingdom, **3** Lydia Becker Institute of Immunology and Inflammation, Wellcome Centre for Cell Matrix Research and Faculty of Biology, Medicine and Health, University of Manchester, Manchester, United Kingdom

☉ These authors contributed equally to this work.
¤a Current address: The Francis Crick Institute, London, United Kingdom
¤b Current address: School of Infection and Immunity, University of Glasgow, Glasgow, United Kingdom
¤c Current address: Cambridge Stem Cell Institute, University of Cambridge, Cambridge, United Kingdom
* Matt.Berriman@glasgow.ac.uk (MB); mad75@cam.ac.uk (MAD-C)

**Data Availability Statement:** The transcriptomic datasets generated during and analysed in the current study are available in the European Nucleotide Archive (ENA) repository (https://www.

## Abstract

Whipworms (*Trichuris* spp) are ubiquitous parasites of humans and domestic and wild mammals that cause chronic disease, considerably impacting human and animal health. Egg hatching is a critical phase in the whipworm life cycle that marks the initiation of infection, with newly hatched larvae rapidly migrating to and invading host intestinal epithelial cells. Hatching is triggered by the host microbiota; however, the physical and chemical interactions between bacteria and whipworm eggs, as well as the bacterial and larval responses that result in the disintegration of the polar plug and larval eclosion, are not completely understood. Here, we examined hatching in the murine whipworm, *Trichuris muris*, and investigated the role of specific bacterial and larval structures and molecules in this process. Using scanning and transmission electron microscopy, we characterised the physical interactions of both fimbriated (*Escherichia coli*, *Salmonella typhimurium* and *Pseudomonas aeruginosa*) and non-fimbriated (*Staphylococcus aureus*) bacteria with the egg polar plugs during the induction/initiation stage, and visualised the effects of structural changes in the polar plugs, leading to larval eclosion. Further, we found that protease inhibitors blocked whipworm hatching induced by both fimbriated and non-fimbriated bacteria in a dose-dependent manner, suggesting the partial involvement of bacterial enzymes in this process. In addition, we identified the minimal egg developmental timing required for whipworm hatching, and transcriptomic analysis of *T. muris* eggs through embryonation revealed the specific upregulation of serine proteases (S01A family) in fully embryonated eggs containing 'hatch-ready' L1 larvae. Finally, we demonstrated that inhibition of serine proteases with the serine-protease inhibitor Pefabloc ablated *T. muris* egg hatching induced by bacteria. Collectively, our findings unravel the temporal and physicochemical bacterial-egg interactions leading to whipworm hatching and indicate serine proteases of both bacterial and larval origin mediate these processes.

ebi.ac.uk/ena/browser/home) under the project accession number PRJEB35377. Sample metadata is available in S1 Table. The full set of GO terms, enrichment scores, and associated transcripts can be visualised using gProfiler with the following links and Supplementary Tables for six week (https://biit.cs.ut.ee/gplink/l/TyiRF6NkQ8; S3 Table) and eight week (https://biit.cs.ut.ee/gplink/l/GkYjEdMISw; S4 Table) over-expressed genes. The R code used to analyse the bulk RNA-seq data of this study is available in the Github repository: https://github.com/Duque-Correa-Lab/Goulding-et-al.-Hatching_Paper_Scripts.

**Funding:** This work was supported by the Sir Henry Dale Fellowship jointly funded by the Wellcome Trust and the Royal Society (222546/Z/21/Z, M.A.D-C.); the Wellcome Trust (206194, M.B; 203151/Z/16/Z, 203151/A/16/Z, M.A.D-C) and the UKRI Medical Research Council (MC_PC_17230, M.A.D-C). SRD is supported by a UKRI Future Leaders Fellowship (MR/T020733/1). The funders had no role in study design, data collection and analysis, decision to publish, or preparation of the manuscript.

**Competing interests:** The authors have declared that no competing interests exist.

## Author summary

Human whipworms are parasites that cause the gastrointestinal disease trichuriasis in millions of people around the world. Infections occur when whipworm eggs, ingested in contaminated food and water, hatch in the intestine in response to gut bacteria (microbiota). The egg encloses a larva within an egg-shell and has a plug at each end. Hatching liberates the larva that burrows inside the cells that line the gut. Interactions between the microbiota of the gut and whipworm eggs are needed for hatching, but are poorly understood. In this study, using the natural mouse whipworm as an infection model, we show that bacteria bind the whipworm egg plugs during the initial stages of hatching, resulting in their degradation and leading to larval exit. We further show that disintegration of the egg plugs is caused by protein-degrading enzymes produced by the bacteria and the larvae. The production of those enzymes by the parasite is dependent on the full development of the larva inside the whipworm egg. These new mechanistic insights pave the way for future studies to understand human whipworm infection and develop new tools to tackle these globally important parasites.

## Introduction

Human whipworm (*Trichuris trichiura*) infects hundreds of millions of people around the globe and can cause the neglected tropical disease Trichuriasis, leading to chronic morbidity with dire socio-economic consequences [1,2]. In addition to the human infective species, over 70 members of the *Trichuris* genus infect a variety of domestic and wild mammals, considerably impacting animal health [2,3]. *Trichuris trichiura* is an obligate parasite of human and non-human primates that cannot be maintained in the laboratory yet and, therefore, the natural whipworm of mice *T. muris* has served as an important model system to study the life cycle and infection pathogenesis of whipworms [2–7].

Whipworm infection occurs upon the ingestion of eggs that hatch in the caecum and proximal colon of their hosts [2,8,9]. Motile first-stage (L1) larvae released from the eggs burrow into the intestinal epithelium, where they grow and moult through the larval (L2-L4) and adult stages [10–13]. The life cycle is completed when females release thousands of eggs a day that are expelled within the host faeces [2,4]. These eggs are un-embryonated and, thus, in a non-infective state; their embryonation occurs in the environment in a process dependent on temperature, oxygen availability, humidity and soil type that takes two to eight weeks and results in the development of the L1 larvae within the egg [2,14–19]. Once embryonated, eggs can remain infective for at least five years, persisting in the environment for long periods and enabling cycles of re-infection that perpetuate infections in endemic areas, representing a significant barrier to the elimination of trichuriasis [16,17,20,21]. Despite the importance of the egg on whipworm transmission, advances in the understanding of *Trichuris* ova development and hatching have been slow in the last four decades [16,22].

The embryonated eggs of whipworms are barrel-shaped and, according to recently proposed nomenclature [23], comprise the following: (1) a Rigid Eggshell Wall, itself consisting of three layers—an outer *Pellicula Ovi* (PO, formerly external vitelline layer), a middle chitinous layer (CL), and an electron-dense parietal coating (EdPC, formerly lipid-rich layer); (2) the Parietal Space (formerly peri-vitelline space) comprising three layers—the *Spatio Externum*, the thin Permeability Barrier Membrane (PBM, formerly the internal vitelline membrane) that encloses the L1 larvae, and the *Spatio Internum;* and (3) a polar or opercular plug at each end

[23–27]. The arrangement of these layers makes the eggs resistant to stress but selectively permeable, enabling them to respond to the outside environment, thereby preventing premature hatching but triggering it under favourable conditions [22,28].

Hatching of *T. muris* eggs is initiated in response to host-gut microbiota at an optimal temperature of 37˚C [8,9]. This leads to the activation of the quiescent L1 larva within the egg, which performs widespread movements followed by localised movements that result in the head penetrating one of the polar plugs [9]. Concomitantly to the larval movements, there is a significant increase in the volume of the egg likely due to endosmosis and an increase in the plug size at the involved pole. The PO over this polar plug resists the mounting pressure until the PBM and the EdPC are pierced by the larval stylet just before eclosion [9,23,27]. Finally, the anterior portion of the L1 larva explosively emerges through the polar canal as the plug ruptures, and then, the rest of the larva slowly squeezes out of the egg through the polar constriction [9].

Several studies suggest that whipworms have evolved to specifically hatch in response to microbiota from or immediately close to the site of infection. Specifically, *T. muris* hatching *in vivo* has been observed in the mouse ileum, caecum and colon [29], and *in vitro* in the presence of murine caecal explants [8] or the contents of the caecum and colon, but not the ileum [30]. The involvement of microbiota in *T. muris* hatching has been confirmed in studies showing that *T. muris* presents a reduced establishment in antibiotic-treated mice [8] and is unable to hatch and colonise germ-free mice [31]. Germ-free mice can sustain an infection, only after colonisation with a faecal transplant from uninfected mice or monocolonisation with *Bacteroides thetaiotaomicron* [31], *Escherichia coli* [32], *Staphylococcus aureus* [33] and *Paraclostridium sordellii* [34]. Consistently, *T. muris* eggs hatch *in vitro* in response to monocultures of Gram-negative (*E. coli*, *Pseudomonas aeruginosa*, *Salmonella enterica* serovar Typhimurium and *Enterobacter hormaechei)* and Gram-positive (*S. aureus*, *Staphylococcus epidermidis*, *Bacillus subtilis*, *P. sordellii*, *Romboutsia hominis*, *Enterococcus caccae* and *Lactobacillus reuteri)* bacteria [8,32–37].

The ability to stimulate hatching of *T. muris* eggs with a single bacteria species *in vitro* has enabled the study of egg-bacterium interactions and molecular mechanisms involved in whipworm egg hatching. Specifically, Hayes and colleagues showed that direct physical contact between intact bacteria (*E. coli*) and *T. muris* eggs is required for hatching, with bacteria clustering around the polar plug of the egg where the larva emerges; an interaction facilitated by type-1 fimbriae [8]. Some of these findings have been recently confirmed using *S. aureus* [33]. Moreover, it has been shown that bacterial metabolic activity is needed for *T. muris* hatching and, in the case of *E. coli* but not *S. aureus*, the hatching depends on arginine biosynthesis [32,33]. However, this disagrees with the previous finding that cultures containing gentamicin-killed bacteria (*E. coli*), where the structural integrity of the bacteria is intact, can induce hatching [8]. Finally, in our preliminary work [38], we have visualised the initial stages of asymmetric degradation of the egg polar plugs upon exposure to *E. coli*. In a recent follow up using *E. coli* and *S. aureus* [33], Robertson and colleagues, suggested that chitinases present in *T. muris* and *T. trichiura* eggs could mediate the disintegration of chitin-protein complexes present in the plug [33]. While some light has been shed on how bacteria promote whipworm hatching, the mechanisms do not seem to be shared across bacterial species. This highlights gaps in our understanding of the physical and molecular interactions between bacteria and whipworm eggs as well as the intrinsic larval factors that facilitate the induction and progression of hatching leading to the degradation of the polar plugs and larval eclosion.

In this study, we investigated temporal, structural and chemical interactions of bacteria (including fimbriated and non-fimbriated and Gram-positive and -negative species) with *T. muris* eggs that trigger and mediate hatching and, examined the influence of larval

development through egg embryonation on hatching readiness. Our results unravel reciprocal effects between the bacteria and the whipworm egg upon binding on the polar plug and uncover the critical role of serine proteases on *T. muris* hatching potentially through the disintegration of the polar plug.

## Materials and methods

### Ethics statement

Mouse experiments were performed under the regulation of the UK Animals Scientific Procedures Act 1986 under the Project licence P77E8A062 and were approved by the Wellcome Sanger Institute Animal Welfare and Ethical Review Body.

### Bacterial culture

The bacteria *E. coli* (strains K12 MG1655 and PK1162 (expressing Green Fluorescent Protein (GFP)—generously provided of Prof. P. Klemm, Technical University of Denmark, Lyngby, Denmark)), *P. aeruginosa* (strains DWW1 [39] and (Schroeter 1872) Migula 1900 (DSM 50071)), *S. aureus* (strains 533 R4 (DSM 20231) and MSSA476 ST1) and *S. enterica* serovar Typhimurium (strain LT2 (DSM 17058); hereafter, *S. typhimurium*) were cultured under standard aerobic conditions (37°C, 5% $CO_2$) in Luria Bertani (LB) medium broth and agar plates. Liquid cultures were shaken at 180 rpm overnight.

### Mice

NSG (NOD.Cg-*Prkdcscid Il2rgtm1Wjl*/SzJ) mice were maintained under specific pathogen-free conditions, under a 12 h light/dark cycle at a temperature of 19–24°C and humidity between 40 and 65%. Mice were fed a regular autoclaved chow diet (LabDiet) and had *ad libitum* access to food and water. All efforts were made to minimise suffering by considerate housing and husbandry. Animal welfare was assessed routinely for all mice involved. Mice were naive prior to the studies described here.

### *Trichuris muris* infections of mice and parasite egg collection

Infection and maintenance of *T. muris* were conducted as previously described [40]. Briefly, NSG mice (6–8 weeks old) were orally infected under anaesthesia with isoflurane with a high dose of embryonated eggs (n ≈ 400 eggs) from the *T. muris* E-isolate [40]. Mice were monitored daily for general condition and weight loss. Thirty-five days later, mice were culled by cervical dislocation and the caecum and proximal colon were removed. The caecum was split and washed in RPMI-1640 plus 500 U/mL penicillin and 500 μg/mL streptomycin (all from Gibco). Worms were removed using fine forceps and cultured for 4 h or overnight in RPMI-1640 plus 500 U/mL penicillin and 500 μg/mL streptomycin at 37°C, 5% $CO_2$. The excretory/secretory (E/S) products from the worm culture were centrifuged (720 *g*, 10 min, room temperature (RT, 20–22°C)) to pellet the eggs. The eggs were allowed to embryonate for eight weeks in distilled water in the dark at RT [31,41], and infectivity was established by worm burden in NSG mice. Upon completion of embryonation, eggs were long-term stored at 4°C, where they maintain their *in vitro* hatching potential and *in vivo* infectivity.

### Characterisation of *T. muris* egg embryonation

Eggs collected from adult parasites recovered from six *T. muris*-infected NSG mice were embryonated separately. During the 12-week embryonation period, an aliquot of ≈ 500 *T. muris* eggs was collected weekly, and larvae development was monitored by light microscopy

using a Zeiss Axio Observer 5 microscope. Hatching in response to *E. coli* was examined on day 0 and during weeks 1, 5, 6, 7, 8 and 12 of embryonation. Gene expression differences were assessed on eggs embryonated for six and eight weeks through RNA sequencing (RNA-seq).

### *In vitro* hatching of *T. muris* eggs with bacteria

*Trichuris muris* eggs were seeded in 96-well plates (Costar) at a density of 1000–3000 eggs per mL (50–150 eggs in 50 μL) and co-cultured with 150 μL of an overnight bacterial culture. Each condition was run in triplicate. Plates were incubated for 2, 4 or 24 h under standard aerobic conditions (37˚C, 5% $CO_2$).

To determine the effects of proteases on hatching, protease inhibitors (Roche) were prepared according to the manufacturer's instructions and added to the eggs and bacteria co-cultures at the following concentrations: cOmplete, Mini, EDTA-free Protease Inhibitor Cocktail 1x / 2x, antipain 50 μg/mL, leupeptin 0.5 μg/mL, Pefabloc SC 1 mg/mL / 2 mg/mL, chymostatin 60 μg/mL, aprotinin 2 μg/mL, and E-64 10 μg/mL.

To investigate if eggs could still be induced to hatch after exposure of co-cultures to the Protease Inhibitor Cocktail for 24 h, eggs were washed twice with sterile water by centrifugation at 720 *g* for 10 min at RT, re-seeded in a 96-well plate (Costar) in 50 μL of water, and incubated with 150 μL of the fresh overnight *E. coli* culture. Plates were incubated for 24 h under standard aerobic conditions (37˚C, 5% $CO_2$).

For all hatching experiments, results were reported as percentage hatching where:

Percentage Hatching = (hatched larvae / [hatched larvae + unhatched embryonated eggs]) x 100

### Live imaging phase contrast and fluorescent microscopy

*Trichuris muris* eggs were co-cultured with GFP-expressing *E. coli*, as described above, for 75 min. Next, co-cultures were pooled and filtered through a 30 μm cell strainer and washed extensively with warm Phosphate Buffered Saline (PBS) 1X. Eggs were then collected in 1 mL of PBS and transferred into a μ-Dish 35 mm, high glass bottom (Ibidi). Images were acquired on a DMI8 Thunder live cell imaging system (Leica) using a 63x/1.4NA apochromatic oil immersion objective lens. The specimens were imaged using both a phase contrast white LED transmitted light image and green epifluorescence, illuminating the sample with a 470 nm LED line with a 535 nm short pass filter in front of the camera to minimise the autofluorescence signal. Eggs were maintained at 37˚C and 5% $CO_2$. The images were collected using a Leica K8 camera and processed using the Leica Application Suite X software. Three independent experiments were performed with at least 20 eggs imaged for each interaction.

### Scanning and transmission electron microscopy

*Trichuris muris* eggs and bacterial strains were co-cultured, as described above, for 60–90 min. Next, co-cultures were fixed by adding an equal volume of solution containing 2% paraformaldehyde (PFA) (Thermo) and 5% glutaraldehyde (GA) (Sigma-Aldrich) in 0.1 M sodium cacodylate buffer (SCB) (Sigma-Aldrich) at pH 7.42 for 30 min at RT. The samples were then transferred to 2 mL round-bottom microfuge tubes in fresh fixative containing 1% PFA and 2.5% GA in SCB and centrifuged at 700 *g* for 10 min at RT.

For scanning electron microscopy (SEM), tubes were left to stand for another 90 min to fix and anchor the eggs vertically in the bottom of the tubes. The eggs were carefully rinsed in SCB three times by centrifugation at 280 *g* for 5 min at RT before post-fixing and layering with 1% osmium tetroxide (Sigma-Aldrich) and thiocarbohydrazide (Sigma-Aldrich) in SCB over 2 h [42]. The eggs were then dehydrated through an ethanol series (one-hour soaks in 20, 30, 50, 70 and 95% ethanol, followed by two one-hour washes in absolute ethanol). The tip of each

tube, containing approximately 200 eggs, was carefully cut off with a razor blade and dried in a Leica EM CPD300. The cut tips were mounted eggs-up onto Hitachi SEM stubs with Leit-Silver (Sigma-Aldrich) before coating with 6 nm of gold in a Leica EM ACE600 evaporation unit and viewed in a Hitachi SU8030 SEM. Four processing runs were performed for co-cultures with *E. coli* to optimise timing and imaging, and two for each of the other bacteria species. Thirty eggs were observed for each interaction.

For transmission electron microscopy (TEM), following primary fixation, the pellet was rinsed carefully with SCB and post-fixed with 1% osmium tetroxide for an hour. Next, the eggs were rinsed three times, and mordanted with 1% tannic acid (Sigma-Aldrich), to stabilise the polar plug during dehydration through an ethanol series performed as above. Eggs were then stained with 1% uranyl acetate *en bloc* at the 30% stage, transferred to propylene oxide for 30 min and embedded in the Epoxy Resin Kit (all from Sigma-Aldrich).

Semi-thin 0.5 μm sections for light microscopy were cut and collected onto clean glass slides. They were dried at 60˚C before staining with 1% Toluidine Blue and 1% Borax (all from Sigma-Aldrich) in distilled water for 30 s. Sections were then rinsed in distilled water, mounted in DPX (Sigma-Aldrich), and coverslipped. Sections were imaged on a Zeiss 200 M Axiovert microscope.

Ultra-thin 60 nm sections were cut using a diamond knife on a Leica EM UC6 ultramicrotome, contrasted again with uranyl acetate and lead citrate and imaged using a TViPs Tem-Cam XF 4.16 on a 120kV FEI Spirit Biotwin TEM. The number of processing runs was the same as above for SEM, with 20 eggs observed for each interaction.

### RNA isolation from *T. muris* eggs and library preparation for RNA-seq

Five hundred *T. muris* eggs collected at six and eight weeks of embryonation were stored in water at -80˚C until RNA extraction. Eggs were thawed on ice, washed three times in ice-cold PBS containing SUPERase•In RNase Inhibitor (Thermo Fisher) at 1 U/μL and pelleted by centrifugation at 20,000 *g* for 2 min at 4˚C. Washed eggs were resuspended in 500 μL of TRIzol LS (Thermo Fisher), placed in lysing matrix D (1.4mm ceramic spheres, MagNA Lyser Green Bead tubes, Roche), and homogenised in a FastPrep24 (MP Biomedicals) at 6.0 m/s for 20 s. Homogenised samples were transferred to 1.5 mL microfuge tubes, and total RNA was extracted by adding 267 μL of chloroform (Sigma-Aldrich), shaking vigorously and incubating for 5 min at RT. Samples were centrifuged at 15,000 *g* for 15 min at RT, and the upper aqueous phase was recovered and mixed with one volume of 100% ethanol. RNA was recovered using the RNA Clean and Concentrator kit (Zymo Research). Total RNA was quantified by Bioanalyzer (Agilent). RNA-seq libraries were constructed using the NEB Ultra II RNA custom kit (New England BioLabs) according to the manufacturer's instructions. The libraries were then pooled in equimolar amounts and sequenced on two lanes of an Illumina HiSeq 4000 using 75 bp paired-end read chemistry (*S1 Table*).

### Bioinformatic analyses

**RNA-seq data processing.** Raw sequencing reads were first assessed using FastQC v.0.12.1 (https://www.bioinformatics.babraham.ac.uk/projects/fastqc/) and MultiQC v.1.17 [43]. Sequencing reads from each RNA-seq sample were pseudo-aligned to the predicted *T. muris* full-length transcripts (PRJEB126.WBPS18.mRNA_transcripts) downloaded from WormBase ParaSite WBPS18 (WS285) [44] using *kallisto* (v.0.46.1) [45]. Briefly, the reference was first prepared using *kallisto index*, after which, data from the two independent lanes per sample were pseudo-aligned to the indexed reference using *kallisto quant* with default parameters and 100 bootstraps.

**Differential expression.**   Data exploration, filtering, and differential gene expression analyses were performed using *DESeq2* (v.1.42.0) [46]. Principal component analyses (PCA) were performed to visualise the sample and replicate distribution. Data were adjusted to log2 (counts) using variance stabilising transformations (VST) before PCA to account for the large differences in log count data when mean counts are low. Before the differential expression analysis, raw counts were filtered to remove low-abundance transcripts (sum of transcripts across all samples <1); this resulted in 16928 transcripts (97.1% of total) retained for analysis.

Differentially expressed transcripts were determined using a Wald test of raw counts between six-week and eight-week sample sets *(S2 Table)*. Transcripts were defined as being significantly differentially expressed based on fold change (log2(FC) >1 or <-1) and a Wald test adjusted p-value (Padj < 0.01). To visualise differentially expressed transcripts, volcano plots were used to compare all transcripts or specific sets of transcripts based on GO term assignment; in both cases, the thresholds described above were applied.

**GO term enrichment.**   Enriched GO terms were identified by g:Profiler (version e111_eg58_p18_30541362, database updated on 25/01/2024) using all known genes as a background and a g:SCS threshold with a significance cutoff of p<0.05. In the Main Figures, the driver GO terms are shown; however, the full set of GO terms, enrichment scores, and associated transcripts can be visualised using gProfiler with the following links and Supplementary Tables for six-week (https://biit.cs.ut.ee/gplink/l/TyiRF6NkQ8; *S3 Table*) and eight-week (https://biit.cs.ut.ee/gplink/l/GkYjEdMISw; *S4 Table*) over-expressed genes.

## Statistical analyses

The Prism 10 software (GraphPad) was used to make statistical comparisons of egg hatch data. Comparisons between two groups (i.e. the absence and presence of protease inhibitor) were performed using Mann–Whitney U two-tailed tests. Kruskal Wallis and Dunn's comparison tests were used for those between three or more groups. RNA-seq data was analysed in *R* (v.4.3.1) as described above.

## Results

### Fimbriated and non-fimbriated bacteria aggregate and interact with the polar plugs and collars of *T. muris* eggs to initiate hatching

Hayes and colleagues previously demonstrated that *T. muris* egg hatching is triggered by both fimbriated (*E. coli*, *S. typhimurium* and *P. aeruginosa*) and non-fimbriated bacteria (*S. aureus*) and showed that *E. coli* clusters at the polar plugs of the eggs using fluorescence microscopy [8]. To better understand the interplay between the bacteria and the eggs that initiate hatching, we visualised the physical interactions of *E. coli*, *S. typhimurium*, *P. aeruginosa* and *S. aureus* with the polar plug of *T. muris* eggs during the early induction stages of hatching (60–90 min of co-culture) using electron microscopy. SEM images revealed that *E. coli* interactions with the polar plugs and collars of the eggs are facilitated by fimbrial projections *(Figs 1A* and *S1A, insets II and III)*. Similarly, co-culture of *T. muris* eggs with *S. typhimurium*, another type-1 fimbriae expressor, resulted in fimbriae-expressing bacteria aggregating around the polar plugs and egg collars *(Fig 1B, insets II and III)*. *P. aeruginosa* is not a type-1 expressor but also localised around the polar plugs and collars of *T. muris* eggs, anchored via projections, which are potentially type IV pili or putative chaperone-usher pathway (Cup) fimbrial structures [47,48] *(Fig 1C, insets II and III)*. In all three cases (*E. coli*, *S. typhimurium* and *P. aeruginosa*), expression of fimbriae or pili appears to be driven by proximity to the polar plugs because bacteria interacting with other "nonpolar" regions of the eggshell did not show such projections

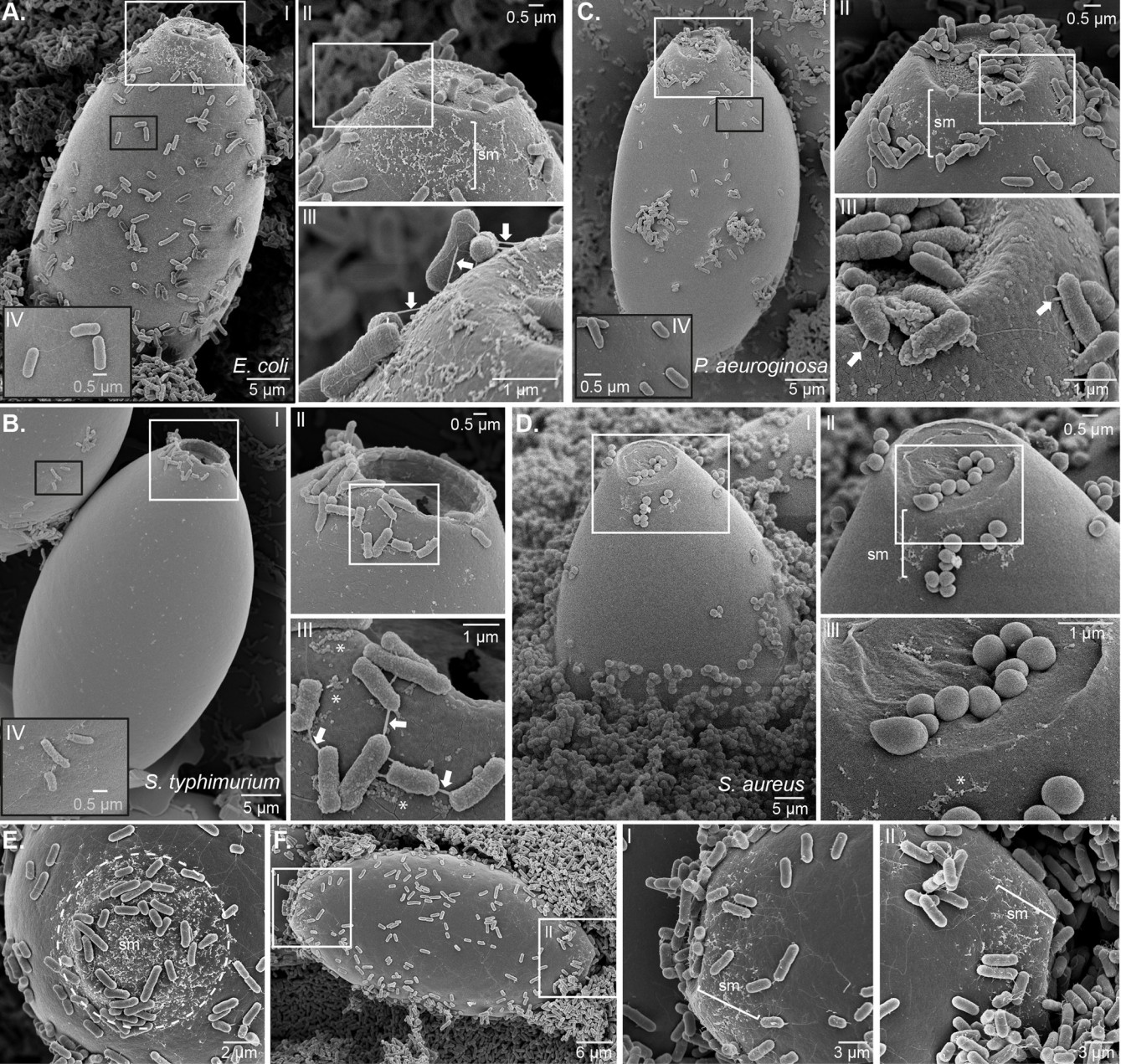

**Fig 1. The aggregation of fimbriated and non-fimbriated bacteria to the polar plugs of *Trichuris muris* eggs precedes hatching.** Representative SEM images from *T. muris* eggs co-cultured for 60–90 min at 37˚C with **A, E** and **F.** *Escherichia coli*, **B.** *Salmonella typhimurium*, **C.** *Pseudomonas aeruginosa*, and **D.** *Staphylococcus aureus*. Fimbriated bacteria (**A, B** and **C**) attach to a zone of surface material (sm) covering the polar plugs of *T. muris* eggs and extending to the egg collars immediately adjacent to the plugs (zoomed into insets II and III, asterisks) via fimbriae/pili (arrows in insets III). Bacteria attached to the eggshell surface in the nonpolar region between the two plugs (black border insets IV) have no fimbriae. Non-fimbriated *S. aureus* (**D**) is in intimate contact with the sm in the collar (asterisk in inset III) and making a cup-like indentation at the surface of the *T. muris* egg polar plug. **E.** The tilted axis view of the polar plug and collar of a *T. muris* egg shows the extent of the sm zone (dashed circle) to which *E. coli* binds. **F.** The sm zone is present at each pole of the eggs, and *E. coli* is visibly binding to both (zoomed insets I and II).

*(Figs 1A–1C, insets IV; S1B, insets II and III).* These results suggest that interactions of fimbriated bacteria with the polar plugs and egg collars of *T. muris* result in the expression of different bacterial extracellular appendages that facilitate bacterial aggregation to the polar plugs

and trigger whipworm egg hatching. Interestingly, the non-fimbriated bacteria *S. aureus* also bound to the polar plugs and collars of *T. muris* eggs; however, these bacteria appear to be in direct intimate contact with the egg, with no appendages observed to mediate this interaction (*Fig 1D*). Whilst *T. muris* egg-hatching is completed by eclosion of the larva through only one of the two polar plugs [9], we observed *E. coli* bacteria binding to a zone of surface material located at both the polar plugs and collars of the egg (*Figs 1E and 1F*). This surface material is also observed in eggs not exposed to bacteria (untreated) (*Fig 2A*).

Next, we expanded these studies with TEM to further visualise the structural and molecular interactions of these bacteria with the polar plugs of *T. muris* eggs. Although preparing samples for ultrastructural TEM leads to shrinking from dehydration of the polar plug and crenulation of the PO [23], clear differences between untreated and bacteria-treated eggs could be seen. The PO of untreated eggs was intact, though wrinkled, and was not bound by any contaminating bacteria in the preparation (*Fig 2B*). In contrast, preparations of eggs treated with *E. coli*, *S. typhimurium* and *P. aeruginosa*, had projections (fimbriae and pili, respectively) directly attaching the bacteria to the PO of the polar plug, and anchoring them to the eggs (*Figs 2C, 2D, 3A and 3B*). As expected, no fimbriae were seen with *S. aureus*; however, the bacterial cell walls caused a shallow cup-like indentation in the PO, indicating an alternative physical interaction with the egg's polar plug (*Fig 3C*). Altogether, these images reveal the different physical interactions between diverse bacteria and the polar plugs and collars of whipworm eggs that occur at the early stages of hatching.

## Temporal and structural changes that result in polar plug degradation and larvae eclosion during bacteria-induced *T. muris* hatching

While the asymmetric disintegration of the polar plugs of *T. muris* eggs that result in hatching has been described [9,33], the temporal and physical (structural) changes in the polar plug layers that lead to their degradation are not known. Here, we harnessed the resolution of TEM to visualise this process when triggered by *E. coli*.

Consistent with previous works [23,24,26,27], unhatched embryonated *T. muris* eggs showed a PO that continuously covered the polar plug and the CL, and their polar plug presented a uniform electron density with no signs of degradation. We also clearly identified the EdPC separating the plug from the *SpEx* and L1 larva (*Fig 4A*). Progression of hatching upon *E. coli* co-culture was accompanied by a striking reduction of electron density within the ILPP, and the EdPC and *SpEx* extending out towards the PO (*Fig 4B*). This was followed by the almost complete disappearance of the plug (with only some of the PO observed by the polar canal) and larva emerging by the polar opening (*Fig 4C, inset I*). However, the hatching egg's contralateral plug remained unbreached, although it presented some degree of degradation (*Fig 4C, inset II*). Interestingly, using SEM, we found bacteria (*S. typhimurium*) still attaching to both polar plugs and collars of the eggs during eclosion (*Fig 4D*), indicating the stability and strength of the interactions of the bacteria with the eggs.

## Progression of hatching of *T. muris* eggs in response to bacterial contact is protease-dependent

Enzymatic degradation of the polar plugs leading to whipworm egg hatching has long been suggested [24,27,49]. While the exact enzyme and its target have not been defined, it was speculated that chitin microfibrils within the polar plugs were susceptible to depolymerisation by chitinases secreted by the larvae [24,27]. This hypothesis was supported by the recent detection of chitinase activity within eggs of both *T. muris* and *T. trichiura* [33]. However, chitin bound

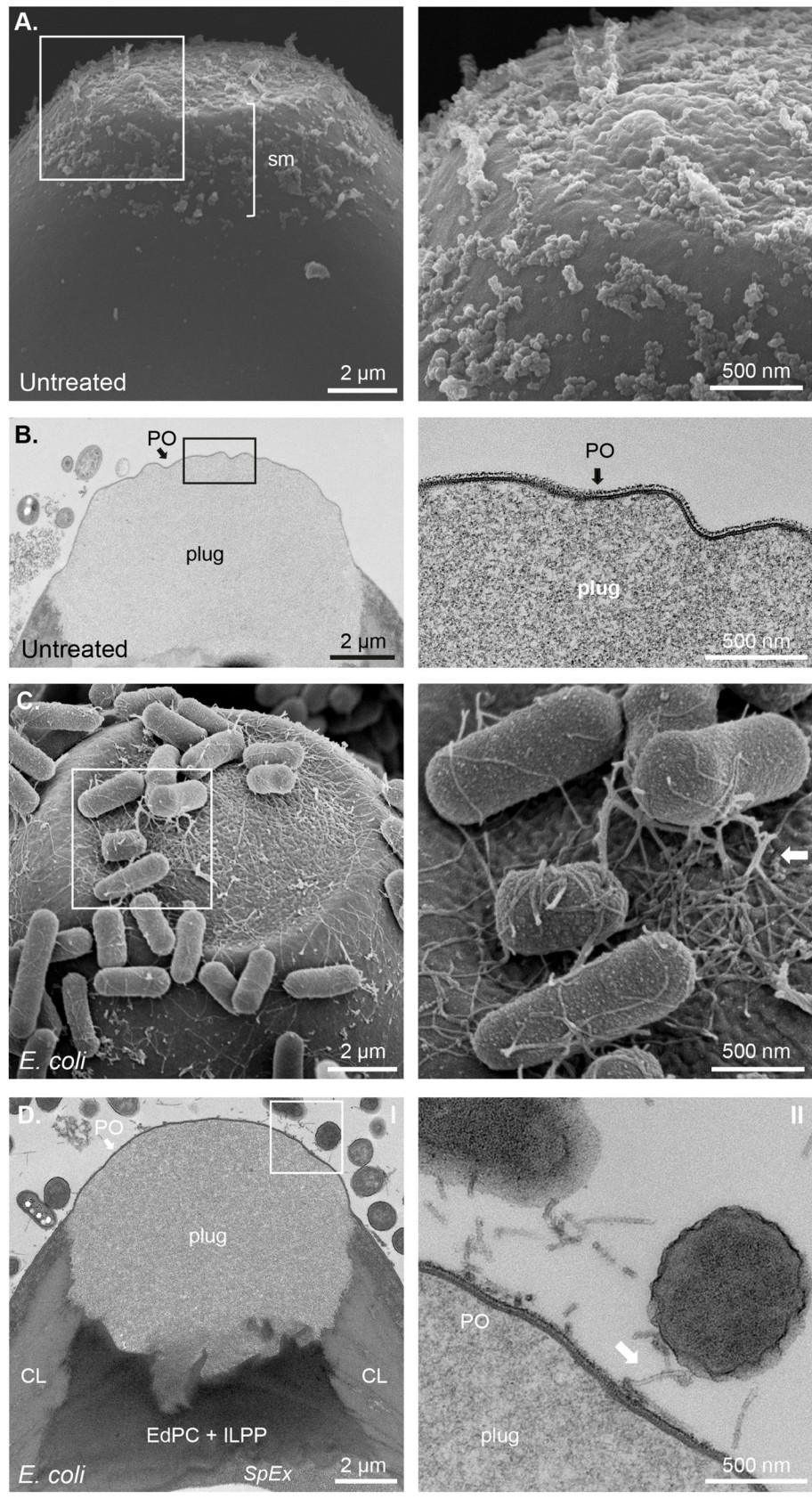

**Fig 2.** *Escherichia coli* anchors to the *Pellicula Ovi* of the polar plugs of *Trichuris muris* eggs via fimbriae during the induction phase of the hatching cascade. Representative SEM and TEM images from *T. muris* eggs untreated (**A** and **B**) or co-cultured with *Escherichia coli* (**C** and **D**) for 90 min at 37°C. Polar plugs and collars of untreated eggs are covered by surface material (sm) detected by SEM (**A**) and display an intact *Pellicula Ovi* (PO) extending across the plug and eggshell (**B**). *E. coli* in contact with the polar plugs and collars of the eggs, express fimbriae (arrows in insets in **C** and **D**) that anchors the bacteria to the PO directly below. Chitin Layer (CL), Electron-dense Parietal Coating (EdPC), Inner Layer of the Polar Plug (ILPP) and *Spatio Externum (SpEx)* are visible in **D**.

to protein, as found in the polar plugs, is not susceptible to degradation by chitinase [24,50,51], suggesting that other enzymes must be driving disintegration of the polar plug.

We hypothesised proteases, but not chitinases, from either bacterial or larval origin, are involved in the degradation of the polar plug. To test this hypothesis, we used a protease inhibitor cocktail comprising a mix of inhibitors with broad activity across most serine and cysteine proteases in our *in vitro T. muris* hatching assays (*Fig 5*). Hatching of *T. muris* eggs induced by both fimbriated and non-fimbriated bacteria was completely ablated in the presence of the protease inhibitor cocktail after 2 and 4 h of co-culture (*Figs 5A and 5B*). Inhibition was shown to be dose-dependent because double the concentration of the protease inhibitor cocktail was required to fully interrupt hatching induced by *S. typhimurium* and *P. aeuroginosa* after 24 h of co-culture, when the bacteria had replicated (*Fig 5C*). We next asked if the action of the protease inhibitor cocktail was irreversible, which could occur if the protease inhibitor cocktail permeated the eggs and affected larval viability or inhibited proteases produced by the larvae. Thus, upon exposure of co-cultures to the cocktail for 24 h, we recovered the eggs, washed them extensively to remove the inhibitors, and re-exposed them to fresh *E. coli* cultures. Hatching capability was restored upon removal of the inhibitor (*Fig 5D*), demonstrating that the protease inhibitor cocktail does not penetrate the *T. muris* eggs and only acts on secreted proteases acting on the outside of the polar plug surface. Collectively, these results indicate that the hatching of whipworm eggs is protease-dependent and suggest that proteases involved in the polar plug degradation can be produced and secreted by the bacteria.

## Complete embryonation of *T. muris* eggs, leading to upregulation of S01A family serine endopeptidases by L1 larvae, is required for hatching

Sufficient development of whipworm L1 larvae may be required to enable the progression and conclusion (eclosion) of egg hatching as larval activation and movement has been shown to mediate the active emergence of the larva from the egg [9]. To better understand the larval developmental and molecular processes involved in hatching, we first identified the minimum timing of *T. muris* egg development required for hatching over an embryonation time course of 8 weeks. We used eggs laid within 24 h by gravid *T. muris* adult females recovered from six NSG mice, which were kept separate as biological replicates. Complete L1 larvae were visible from five weeks of embryonation (*Fig 6A*); however, we only observed hatching in response to *E. coli* after seven weeks of embryonation (*Fig 6B*). After this time, hatching decreased, then stabilised after eight weeks of embryonation (*Fig 6B*). Strikingly, the stage of embryonation of *T. muris* eggs did not affect bacterial binding to their polar plugs and collars (*S2 Fig*). Un-embryonated eggs freshly laid by adult females (*S2B Fig*) and those that remained un-embryonated after 12 weeks of embryonation (*S2C Fig*) showed similar numbers of bacteria binding to both poles of fully developed (embryonated) eggs (*S2A Fig*).

Next, we investigated gene expression changes associated with hatching readiness by comparing the transcriptome of eggs after six and eight weeks of embryonation using RNA-seq. Despite sharing similar morphology (*Fig 6A*), the transcriptomes of 6- and 8-week embryonated eggs differed significantly (*Figs 6C and 6D* and S2 Table). The less mature eggs,

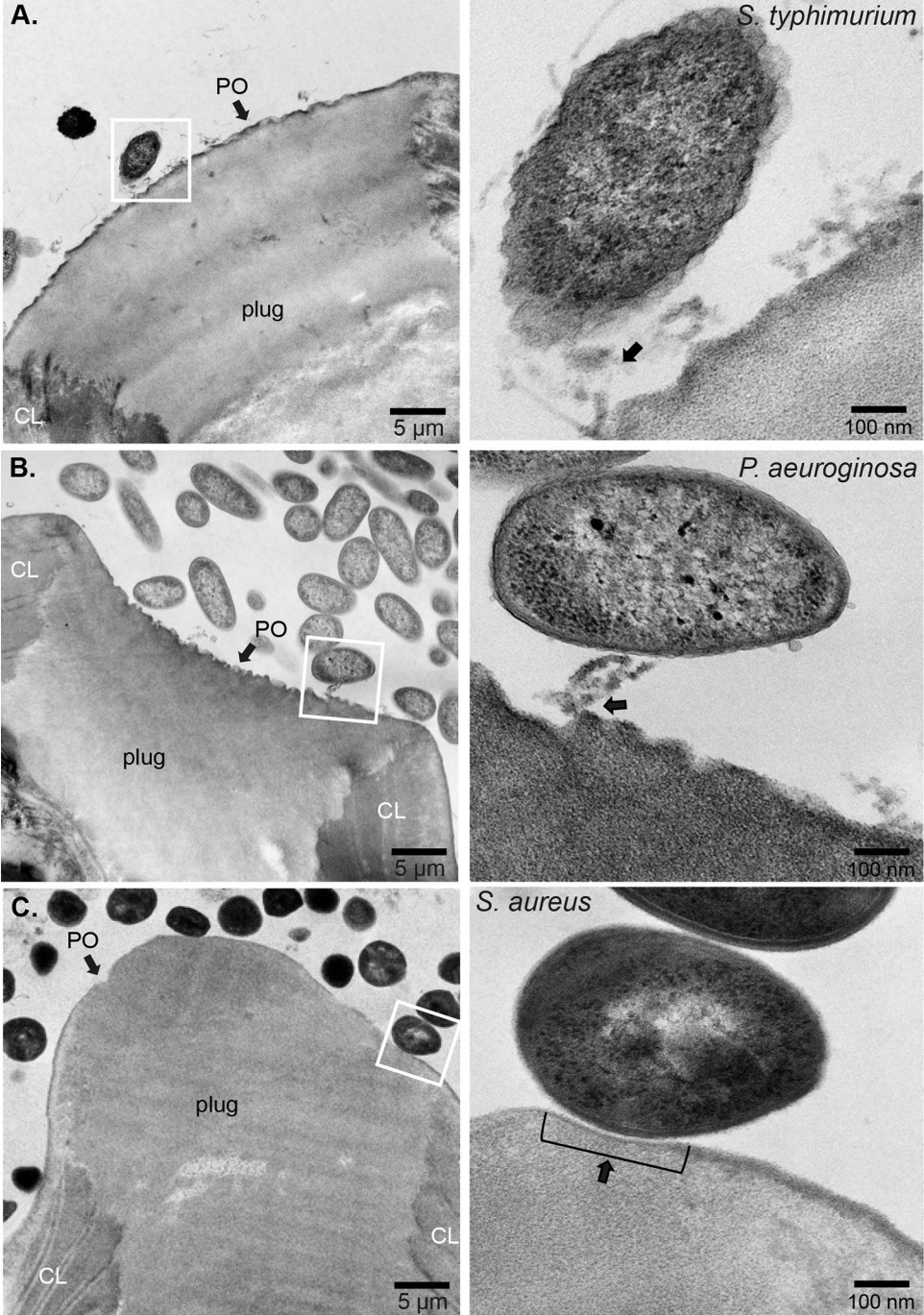

**Fig 3. Interactions of fimbriated and non-fimbriated bacteria with the *Pellicula Ovi* of *Trichuris muris* eggs polar plugs during the induction phase of the hatching cascade.** Representative TEM images from *T. muris* eggs co-cultured for 90 min at 37°C with: **A.** *Salmonella typhimurium*, **B.** *Pseudomonas aeruginosa*, and **C.** *Staphylococcus aureus*, showing bacteria in contact with *Pellicula Ovi* (PO) of the *T. muris* egg polar plug bordering the eggshell chitin layer (CL). Insets show the detail of bacterial attachment to the PO via fimbriae and pili (arrows in insets of **A** and **B**) or direct contact of the *S. aureus* cell wall to a shallow cup-like indentation zone on the PO surface (arrow in inset **C**).

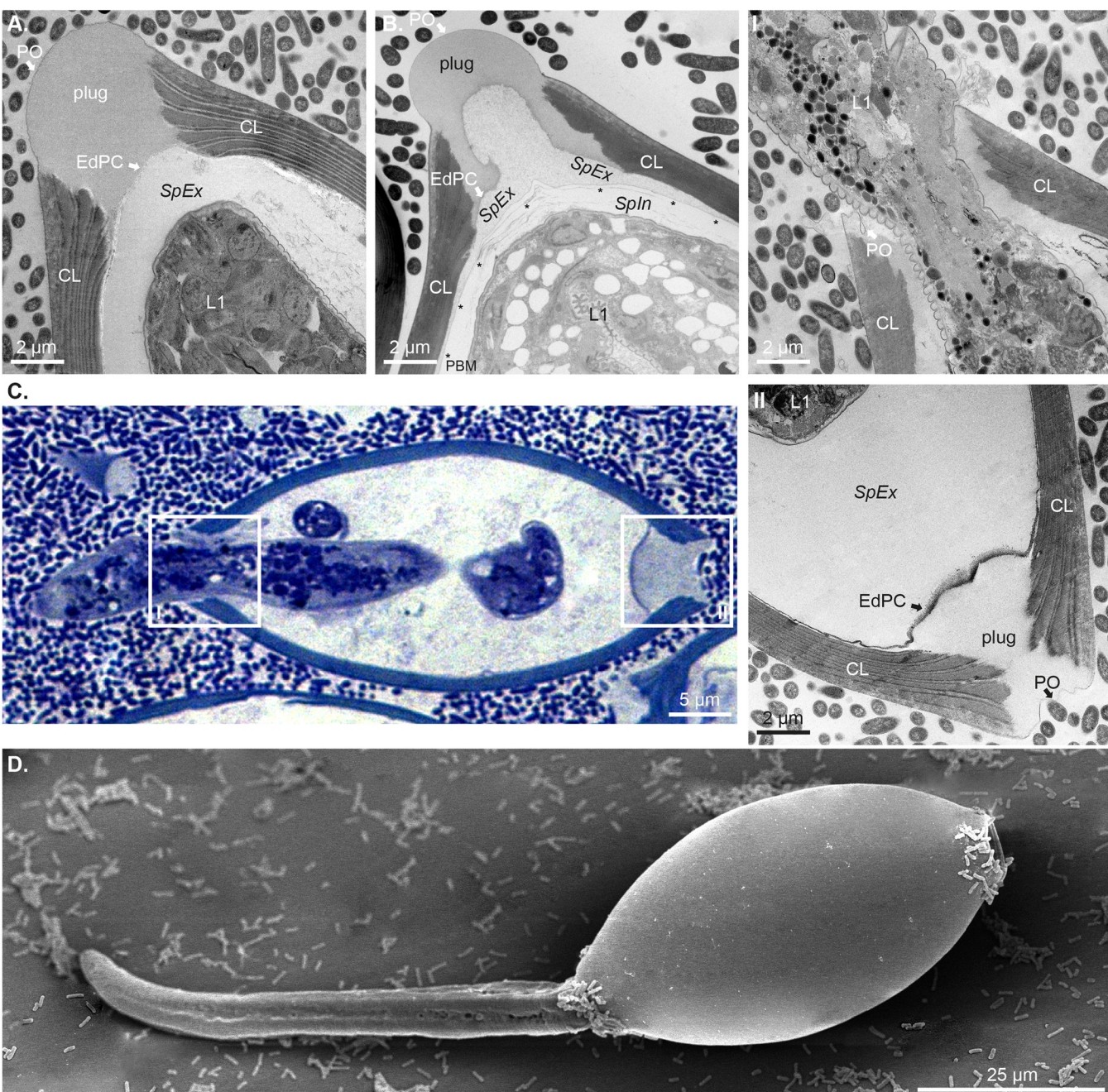

**Fig 4. Temporal and physical changes leading to the asymmetric disintegration of the polar plugs during the hatching of *Trichuris muris* eggs. A-C.** Representative TEM and Toluidine Blue-stained images showing progressive degradation of the polar plug of *T. muris* eggs during hatching induced by *Escherichia coli* at 75–90 min of co-culture. **A.** Intact plug of an embryonated egg showing *Pellicula Ovi* (PO) continuous across the chitin layer (CL) and plug, Electron-dense Parietal Coating (EdPC) separating the plug from the *Spatio Externum (SpEx)* and first-stage larva (L1). **B.** Degradation proceeds with visible loss of electron density within the Inner Layer of the Polar Plug (ILPP), which leads to the EdPC and *SpEx* expanding upwards. L1 larva is enclosed by the Permeability Barrier Membrane (PBM, asterisks) and the *Spatio Internum (SpIn)* is visible. **C.** Finally, one of the two polar plugs is destroyed following degradation and larval eclosion (inset I), while the contralateral polar plug remains unbreached (inset II). **D.** Representative SEM image showing *S. typhimurium* binding both poles of a *T. muris* egg undergoing eclosion.

embryonated for six weeks, expressed genes associated with metallopeptidase activity and growth and development, indicated by GO terms such as 'anatomical structure development', 'structural constituent of cuticle' and 'chitin binding' *(Fig 6E and S3 Table)*. Gene expression

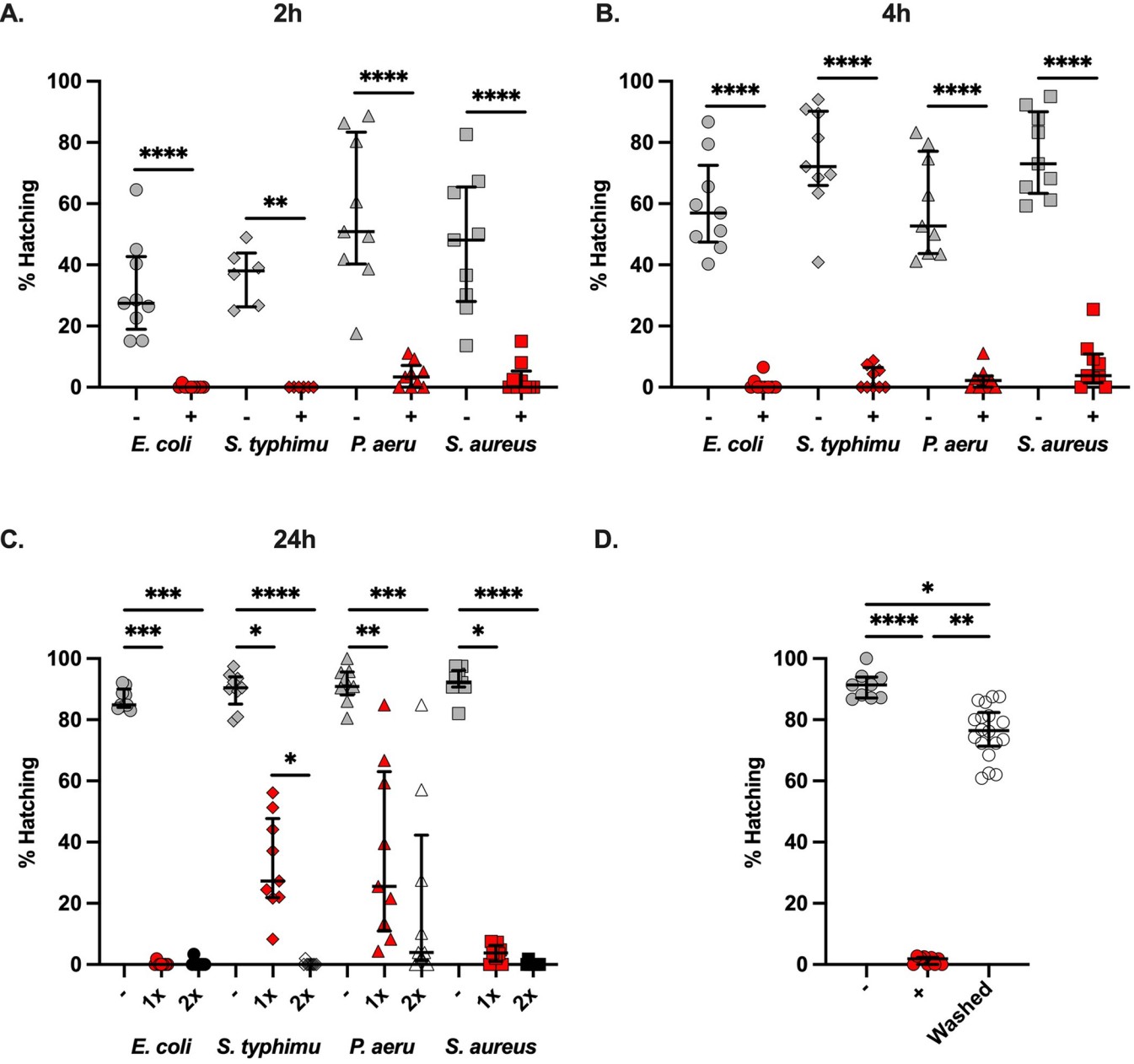

**Fig 5. Bacteria-mediated hatching of *Trichuris muris* eggs is protease-dependent.** *Trichuris muris* eggs were co-cultured with *Escherichia coli*, *Salmonella typhimurium*, *Pseudomonas aeruginosa* and *Staphylococcus aureus* in the absence (-) or presence (+) of a protease inhibitor cocktail (1x) for **A.** 2 h and **B.** 4 h at 37°C. The number of total embryonated eggs and hatched larvae were counted, from which the hatching percentage was calculated. Hatching was completed in triplicate across three independent experiments (n = 9), except for *S. typhimurium* in **A,** where data comes from two independent experiments (n = 6). Median and interquartile range are shown, and statistical differences between the absence (-) or presence (+) of a protease inhibitor cocktail for each bacterial species were evaluated using the Mann-Whitney test (**p<0.005, ****p<0.0001). **C.** *T. muris* eggs were co-cultured with *E. coli*, *S. typhimurium*, *P. aeruginosa* and *S. aureus* in the absence (-) or presence (1x or 2x) of a protease inhibitor cocktail for 24 h at 37°C. The number of total embryonated eggs and hatched larvae were counted, from which the hatching percentage was calculated. Hatching was completed in triplicate across three independent experiments (n = 9). Median and interquartile range are shown, and statistical differences between the absence (-) or presence (1x or 2x) of a protease inhibitor cocktail for each bacterial species were evaluated using Kruskal Wallis with Dunn's multiple comparison tests performed (*p<0.05, **p<0.005, ***p<0.0005, ****p<0.0001). **D.** *T. muris* eggs were co-cultured with *E. coli* in the absence (-) or presence (+) of a protease inhibitor cocktail (1x) for 24 h at 37°C. The number of total embryonated eggs and hatched larvae were counted, from which the hatching percentage was calculated. Next, *T. muris* eggs co-cultured in the presence of the inhibitor were washed to remove it and re-exposed to fresh *E. coli* cultures for 24 h at 37°C; the percentage hatching was recorded 24 h later (Washed). Hatching was completed in triplicate in the absence and presence of the protease inhibitor cocktail and sextuplicate for the washed condition across three independent experiments (n = 9 and n = 18, respectively). Median and interquartile range are shown, and statistical differences between groups were evaluated using Kruskal Wallis with Dunn's multiple comparison tests performed (*p<0.05, **p<0.005, ****p<0.0001).

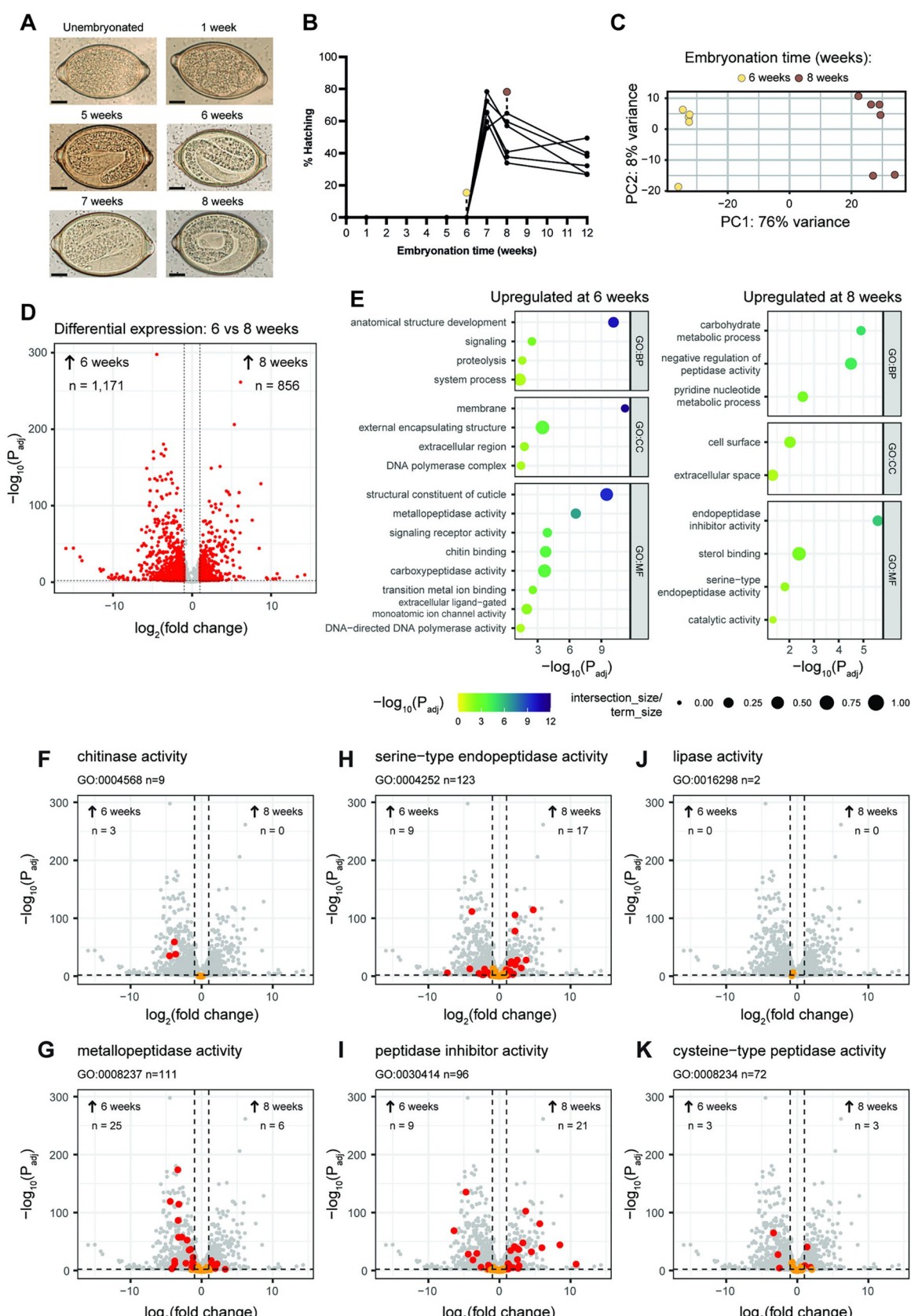

**Fig 6. Complete *Trichuris muris* egg embryonation leads to the development of 'hatching-ready' L1 larvae expressing S01A family serine endopeptidases and inhibitors. A-B.** *T. muris* un-embryonated eggs were placed in deionised water in the dark at room temperature. Embryonation was monitored over time at 0, 1, 5, 6, 7, 8 and 12 weeks by **A.** brightfield imaging (scale bars = 10 μm) and **B.** percentage of egg hatching was determined upon co-culture with *Escherichia coli* for 24 h at 37°C (n = 6). **C-K.** Differential gene expression by DESeq2 between *T. muris* eggs embryonated for six (n = 5) and eight weeks (n = 6). **C.** Principal Component Analysis (PCA) of normalised transcripts (log2, variance stabilised transcripts (VST)), showing the relationship between 6 and 8 week samples and replicates. **D.** Volcano plot of differentially expressed transcripts from *T. muris* eggs embryonated for six (left side) and eight weeks (right side). Dashed lines represent thresholds for fold-change (<-1 and >1) and adjusted p-value (<0.01); transcripts that are significantly differentially expressed are indicated by red points. **E.** Gene set enrichment analysis of transcripts significantly differentially expressed and upregulated at six or eight weeks of embryonation. Gene Ontology (GO) terms were analysed in gProfiler; shown are -log10($P_{adj}$) (colour scale) and term size (size) for enriched GO terms in each time point, categorised by biological process (BP), cellular component (CC), and molecular function (MF). **F-K.** Volcano plots focused on transcripts for specific GO terms. Each plot is assembled as in D; however, transcripts associated with a specific GO term are highlighted, with significantly differentially expressed transcripts (dashed lines indicated thresholds, as in panel D) shown in red, whereas non-significant transcripts are marked in orange. **F.** Chitinase activity (GO:0004568). **G.** Metallopeptidase activity (GO:0008237). **H.** Serine-type endopeptidase activity (GO:0004252). **I.** Peptidase inhibitor activity (GO:0030414). **J.** Lipase activity (GO:0016298). **K.** Cysteine-type peptidase activity (GO:0008234).

in more mature eggs, embryonated for eight weeks, was enriched for GO terms that included 'carbohydrate metabolic process'—reflecting both glycan remodelling and increased glycolysis and related pathways of a more energetically active parasite. Genes with 'serine-type endopeptidase activity', 'endopeptidase inhibitor activity' and 'negative regulation of peptidase activity' were also overrepresented *(Fig 6E and S4 Table)*, suggesting that both peptidase activity and inhibition are major features of 'hatching-ready' L1 larvae.

Considering the involvement of proteases *(Fig 5)*, as well as other enzymes previously proposed such as chitinases and lipases [22,27,33,52], in *T. muris* egg hatching, we next screened the expression of specific groups of genes coding for enzymes and their inhibitors *(Figs 6F–6K and S5–S10 Tables)*. We found the expression of chitinases *(Fig 6F and S5 Table)* and metallopeptidases *(Fig 6G and S6 Table)* to be upregulated in 6-week embryonated eggs, while almost double the number of serine endopeptidases were overexpressed in eggs embryonated for eight weeks relative to those embryonated for six weeks *(Fig 6H and S7 Table)*. Interestingly, the majority of the serine endopeptidases expressed by 8-week embryonated eggs belong to the S01A family *(S7 Table)*. Other enzymes, such as lipases *(Fig 6J and S8 Table)* and cysteine peptidases *(Fig 6K and S9 Table)*, were either not significantly differentially expressed or were not skewed to either time-point. Protease inhibitors were differentially expressed on eggs embryonated for six and eight weeks *(Fig 6I and S10 Table)*. About half of those expressed by eight-week embryonated eggs were serine endopeptidase inhibitors; in contrast, none expressed after six weeks of embryonation belonged to this class *(S10 Table)*. These data suggest that full embryonation of *T. muris* is required to develop 'hatching-ready' L1 larvae expressing S01 family serine peptidases that potentially drive hatching processes from within the egg through a tightly regulated process.

## Hatching of *T. muris* eggs induced by bacterial contact is serine protease-dependent

Our transcriptomic experiments indicate that serine proteases are expressed in hatch-ready L1 larvae, leading us to hypothesise that serine proteases are critical for hatching whipworm eggs. We next used a panel of serine and cysteine inhibitors in our *in vitro T. muris* hatching assays to test this hypothesis and validate our transcriptomic results. Specifically, we tested dual serine and cysteine protease inhibitors with a narrower activity spectrum (antipain and leupeptin) and single inhibitors for both serine (Pefabloc, chymostatin and aprotinin) and cysteine (E64) proteases *(Fig 7A)*. Strikingly, we observed complete ablation of *T. muris* hatching induced by *E. coli* with the serine protease inhibitor Pefabloc, matching that of the protease inhibitor

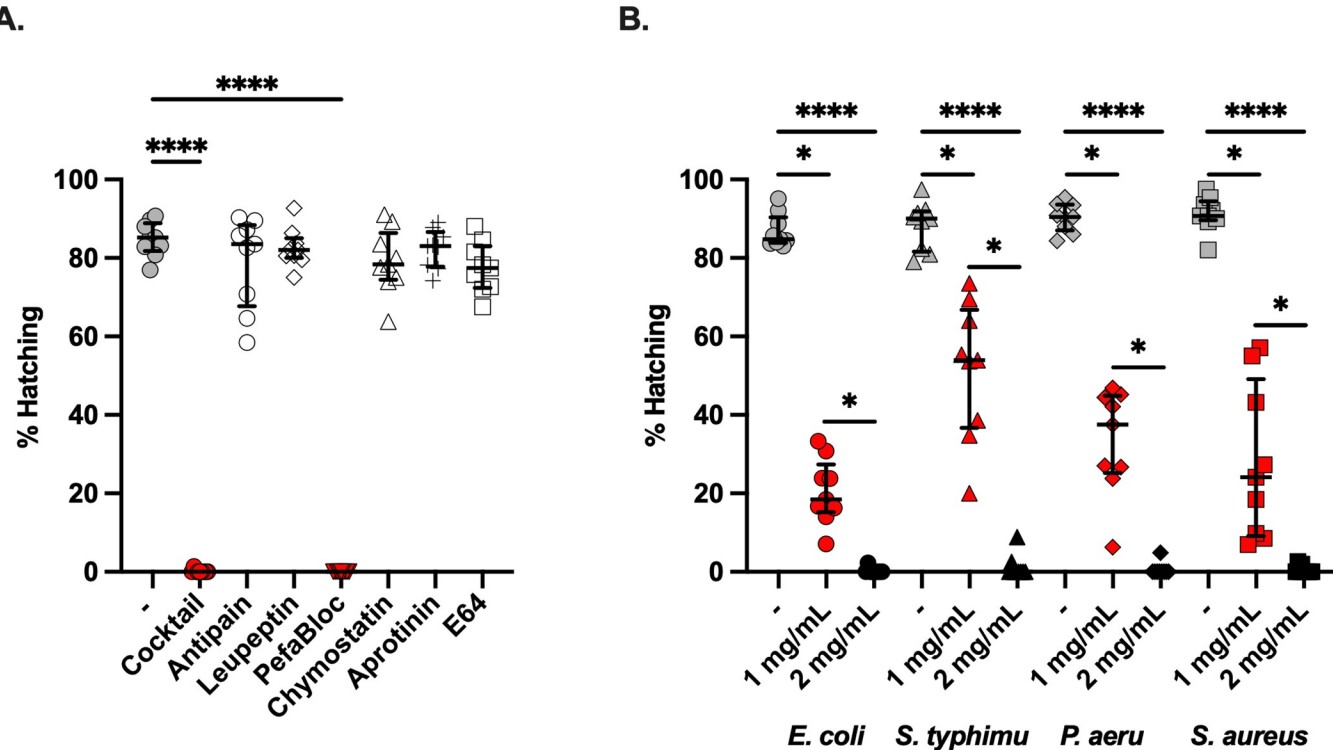

**Fig 7. Bacterial-mediated hatching of *Trichuris muris* eggs is serine-protease dependent. A.** *T. muris* eggs were co-cultured with *Escherichia coli* in the absence (-) or presence of a protease inhibitor cocktail, the dual-class protease inhibitors antipain and leupeptin, the serine protease inhibitors Pefabloc, chymostatin and aprotinin, and the cysteine protease inhibitor E64. Percentage hatching was calculated after 24 h of co-culture at 37°C. Hatching was completed in triplicate across three independent experiments (n = 9). The median and interquartile range are shown. Kruskal Wallis with Dunn's multiple comparison tests were performed (****p<0.0001). **B.** *T. muris* eggs were co-cultured with *E. coli*, *Salmonella typhimurium*, *Pseudomonas aeruginosa* and *Staphylococcus aureus* in the absence (-) or presence (1 mg/mL or 2 mg/mL) of Pefabloc for 24 h at 37°C. The number of total embryonated eggs and hatched larvae were counted, from which the hatching percentage was calculated. Hatching was completed in triplicate across three independent experiments (n = 9). Median and interquartile range are shown, and statistical differences between absence (-) or presence (1 mg/mL or 2 mg/mL) of Pefabloc for each bacterial species were evaluated using Kruskal Wallis with Dunn's multiple comparison tests were performed (*p<0.05, ****p<0.0001).

cocktail. None of the other inhibitors affected *T. muris* hatching (*Fig 7A*). Next, we examined if Pefabloc also inhibits *T. muris* hatching induced by other bacteria. At 24 h of co-culture, Pefabloc ablated the hatching induced by both fimbriated and non-fimbriated bacteria in a dose-dependent manner (*Fig 7B*). These findings demonstrate that serine-proteases, from either bacterial or larval origin, are critically important for hatching *T. muris* eggs.

## Discussion

Hatching is a critical phase of the whipworm life cycle, marking the transition from the "obligatory quiescence" of the embryonated egg to the initiation of the successful establishment of whipworm L1 larvae in their host [9]. It has previously been proposed that *T. muris* hatching is a two-phase event involving (i) the enzymatic hydrolysis of the polar plug material and (ii) the physical rupture of the "lipid" and "vitelline" layers [27]. Our findings, supported by complementary studies [8,33], indicate that the hatching of *T. muris* is a three-phase event, involving (i) physical interactions of bacteria with the egg's polar plugs that precede and trigger hatching, (ii) asymmetrical enzymatic degradation of one polar plug by serine-proteases of both bacterial and larval origin concomitant with larval activation, and (iii) the physical rupture of the EdPC and PBM layers and the involved polar plug by the larvae, which leads to eclosion.

In the first phase, we observed the aggregation of bacteria at the polar plugs of *T. muris* eggs, consistent with previous works [8], and extended these findings using SEM and TEM, revealing the binding of bacteria to surface material present in both polar plugs and collars of the eggs through surface appendages in the case of fimbriated bacteria or by direct contact in the case of *S. aureus (Figs 1–3 and S1)*. Interestingly, our images show a location-specific response by the bacteria, whereby binding to the egg poles results in increased expression of type-1 fimbriae by *E. coli* and *S. typhimurium* and pilli by *P. aeruginosa (Figs 1 and S1)*, potentially driven by chemotaxis to and/or sensing of the surface material at the egg plugs and collars. This response could facilitate bacterial adhesion to the egg as observed during urinary tract infections where attachment of *E. coli* to the uroepithelium results in and is facilitated by increased expression of type-1 fimbriae [53]. Similarly, early attachment of *P. aeruginosa* to surfaces results in overaccumulation of outer membrane and biofilm formation proteins that can support bacterial colonisation [54]. While *S. aureus* binding to the polar plug is not mediated by appendages, cell wall-anchored proteins that contribute to adhesion to and invasion of host cells and tissues and biofilm formation [55] could be implicated in the direct interactions of *S. aureus* with the polar plugs of *T. muris* eggs and warrant future investigations to identify the molecules involved in the induction of hatching by non-fimbriated bacteria. Our observations suggest that increased bacterial aggregation to the polar plug could be the rate-limiting factor in activating downstream processes, including polar plug degradation and larval activation.

During the second-phase of hatching, our images revealed the structural changes on the egg polar plugs that follow *E. coli* binding and that lead to the egg polar plug disintegration and larvae eclosion (*Fig 4*). We first described these results in 2022 [38] and since then, Robertson and colleagues [33] used SEM and serial block face (SBF)-SEM to similarly show bacterial contact with the polar plugs during hatching and visualise the initial stages of the polar plug asymmetric disintegration by *E. coli* and *S. aureus*. In addition, the authors quantified the number of bacteria associated with each plug using an *S. aureus* strain expressing GFP and fluorescence microscopy and showed that bacteria binding to the poles was strikingly asymmetric. In contrast, our SEM and fluorescence images using GFP-expressing *E. coli* showed bacteria interacting with surface material at both polar plugs and collars in similar numbers (*Figs 1F and S2*). Interestingly, this interaction occurs even with un-embryonated eggs (*S2 Fig*) suggesting that the surface material covering the polar regions of the eggs is produced *in utero* and does not change during the development of the larvae inside the egg. Further work is needed to better understand the nature of the surface material on the eggs to which the bacteria bind and the drivers of the asymmetric degradation of the polar plugs.

Our preliminary work [38] and the results presented here, demonstrated that the hatching of *T. muris* eggs induced by both fimbriated and non-fimbriated bacteria was dependent on the activity of secreted proteases and was ablated by protease inhibitors with broad activity for serine and cysteine proteases. These results were confirmed by Robertson and colleagues, who used our treatment approach with a protease inhibitor cocktail to block *T. muris* hatching in response to *S. aureus* [33]. They argue that the degradative enzymes enabling this process (they propose chitinases) are produced and released specifically by the larvae, as it is unlikely that taxonomically-distinct bacteria species possess the same enzyme required to dissolve the polar plug [33]. In contrast, our data clearly indicate the involvement of bacterial proteases, as shown by the dose-dependent action of the inhibitor cocktail associated with the higher bacterial load due to longer incubation and growth time (*Fig 5*).

Our findings also suggest a contribution of larval-derived proteases on hatching, the expression of which increases with larval development through embryonation (*Fig 6*). Complete embryonation of *T. muris* has been defined as the appearance of a fully developed L1 larva

within the egg, the speed of which is affected by temperature, taking three weeks of incubation at 26˚C and six weeks at RT [16]. We show that although fully developed larvae are visible from five weeks post-incubation at RT, bacteria-induced *in vitro* hatching does not occur until eggs have undergone seven weeks of embryonation. Correspondingly, Wakelin reported that full infectivity of eggs in mice is not reached until some time after growth and development had apparently been complete [29], and similar observations have been described for *Ascaris suum* [56]. Intriguingly, we observed a decrease in the egg's hatching potential after eight weeks of embryonation, which may be explained by the transition of larvae to a quiescence state evidenced by a reduction of larval movement within the egg [16].

By comparing the transcriptome of eggs embryonated for six and eight weeks, gene expression changes associated with developmental processes and enzyme activities contributing to hatching readiness were revealed (*Fig 6*). Specifically, we found high expression of metallopeptidases and chitinases in six-week embryonated eggs but no further upregulation during embryonation. The expression of chitinases is correlated with processes of anatomical structure development and the structural constituent of the cuticle, suggesting the role of chitinases in the control of chitin deposition on the larval cuticle during development. Chitinase activity has been shown to increase with the development of eggs of *Heligmosomoides polygyrus*, *A. suum* and *T. muris* [33,57,58], and due to the presence of chitin in the eggshell of these parasites, larval chitinases have been suggested before to contribute to hatching. However, the involvement of larval chitinase activity in hatching has only been directly assessed in *H. polygyrus* using the specific chitinase inhibitor allosamidin, which led to a slower rate of hatching but did not prevent it [58]. Moreover, it has been recently shown in mice that *A. suum* eggs hatch in the host stomach, with the eggshell being degraded by the acidic mammalian chitinase from the gastric epithelium, a process disrupted by chitinase inhibition and the loss of gastric chief and parietal cells that produce this enzyme and maintain acidic conditions required for its activity, respectively [59]. Like *Ascaris*, *Trichuris* is transmitted by oral ingestion of eggs; thus, if chitinases would similarly effectively degrade *Trichuris* egg polar plugs, their passage through the stomach should result in hatching in this organ instead of the caecum. Collectively, our results and these observations suggest that chitinases do not play a major role in the hatching of *Trichuris* eggs and that the increase in their expression during larval development may instead play roles in the formation of the parasite cuticle. In contrast and in keeping with our findings on the critical role of proteases on *T. muris* hatching, complete embryonation of the eggs that led to hatching readiness resulted in an overexpression of serine endopeptidases belonging to the S01 family. Total ablation of *T. muris* hatching induced by both fimbriated and non-fimbriated bacteria by Pefabloc (*Fig 7*) [60,61], an inhibitor of S01 family serine proteases, conclusively demonstrated that serine proteases from either bacterial or larval origin, are responsible for the hatching of *T. muris* eggs.

Building on our results, we hypothesise that binding of bacteria to the polar plug results in secretion of bacterial proteases that initiate the disintegration of the polar plug from the outside. This initial degradation could result in endosmosis that may rehydrate and activate the larvae [9,62] (increased larval activity was evidenced by expression of the glycolysis pathway) to secrete serine proteases that can finalise the degradation of the polar plug from within the egg. Future work is needed to test this hypothesis, identify the specific serine proteases from bacteria and larvae responsible for hatching and unravel the processes controlling their secretion, spatial localisation, activation and action over still unidentified substrates on the polar plugs. Nonetheless, it is also possible that the upregulation of these genes in the parasite plays no role in hatching *per se*, and is instead an adaptation of the parasite to be 'primed' to deal with the external environment or invade the intestinal epithelium once the larvae emerges from the egg.

In summary, this study has uncovered new components of the bacterial-whipworm interactions and key molecular processes mediating *T. muris* egg development and hatching. This new knowledge will steer future research aiming to discover the specific human and non-human primate microbiota-egg interactions controlling the hatching of *T. trichiura*, which could shed light on the host specificity of *Trichuris* species and their co-evolution with their hosts. Moreover, armed with this knowledge, it may be possible to develop *in vivo* and *in vitro* models of human whipworm infection, and identify new intervention targets based on inducing suicidal hatching or inhibiting the hatching cascade to prevent re-infection.

## Supporting information

**S1 Fig. Interactions with the polar plugs and collars of *Trichuris muris* eggs drive the expression of type-1 fimbriae in *Escherichia coli*.** Representative SEM images from *T. muris* eggs co-cultured with *E. coli* for 60–90 min at 37°C. Bacteria attached to the eggshell surface are shown: (**A**) at the polar plug and egg collar; and (**B**) between the two collars. Insets II and III show increased magnification views of bacteria that bind via fimbriae (in **A**) to surface material (sm, asterisks) covering the polar plugs and egg collars. The sm is absent between the two collars (**B**) and bacteria attaching to these areas have no fimbriae.
(TIF)

**S2 Fig. Bacteria binding to polar plugs and collars of *Trichuris muris* eggs is independent of their stage of embryonation.** Representative extended depth of field projections of phase contrast and green fluorescent microscopy z-stack images from *T. muris* eggs—**A.** embryonated, **B.** un-embryonated, freshly laid, and **C.** un-embryonated, after embryonation period—and co-cultured with GFP-expressing *Escherichia coli* for 75 min at 37°C (scale bars = 20 μm).
(TIF)

**S1 Table. Sample metadata for RNA-seq samples.** RNA-seq libraries metadata including sample (original and Sanger) ID and accession number, time post-infection, replicate ID and lane names.
(XLSX)

**S2 Table. Differential transcript expression between six and eight week eggs using DESeq2.** Complete list of differentially expressed transcripts from *Trichuris muris* eggs embryonated for six and eight weeks. Genes were defined as being differentially expressed based on fold-change ($\log_2$ FC $<-1$ and $>1$) and adjusted p-value ($<0.01$).
(XLSX)

**S3 Table. Six-week enriched GOterms using gProfiler.** Gene set enrichment analysis of transcripts significantly differentially expressed and upregulated at six weeks of embryonation. Gene Ontology (GO) terms were analysed in gProfiler; shown are enriched GO term names and IDs, adjusted p-value ($<0.01$), and term and intersection size, categorised by biological process (BP), cellular component (CC), and molecular function (MF).
(XLSX)

**S4 Table. Eight-week enriched GOterms using gProfiler.** Gene set enrichment analysis of transcripts significantly differentially expressed and upregulated at eight weeks of embryonation. Gene Ontology (GO) terms were analysed in gProfiler; shown are enriched GO term names and IDs, adjusted p-value ($<0.01$), and term and intersection size, categorised by biological process (BP), cellular component (CC), and molecular function (MF).
(XLSX)

**S5 Table. Differential expression of transcripts with GOterm chitinase activity.** List of differentially expressed transcripts associated with the Gene Ontology (GO) term Chitinase activity (GO:0004568) from *Trichuris muris* eggs embryonated for six and eight weeks. Gene ID, mean expression, log$_2$ fold-change and adjusted p-value are shown.
(XLSX)

**S6 Table. Differential expression of transcripts with GOterm metallopeptidase activity.** List of differentially expressed transcripts associated with the Gene Ontology (GO) term Metallopeptidase activity (GO:0008237) from *Trichuris muris* eggs embryonated for six and eight weeks. Gene ID, mean expression, log$_2$ fold-change and adjusted p-value are shown.
(XLSX)

**S7 Table. Differential expression of transcripts with GOterm serine-type endopeptidase activity.** List of differentially expressed transcripts associated with the Gene Ontology (GO) term Serine-type endopeptidase activity (GO:0004252) from *Trichuris muris* eggs embryonated for six and eight weeks. Gene ID, mean expression, log$_2$ fold-change, adjusted p-value and MEROPS family and clan are shown.
(XLSX)

**S8 Table. Differential expression of transcripts with GOterm lipase activity.** List of differentially expressed transcripts associated with the Gene Ontology (GO) term Lipase activity (GO:0016298) from *Trichuris muris* eggs embryonated for six and eight weeks. Gene ID, mean expression, log$_2$ fold-change and adjusted p-value are shown.
(XLSX)

**S9 Table. Differential expression of transcripts with GOterm cysteine-type peptidase activity.** List of differentially expressed transcripts associated with the Gene Ontology (GO) term Cysteine-type peptidase activity (GO:0008234) from *Trichuris muris* eggs embryonated for six and eight weeks. Gene ID, mean expression, log$_2$ fold-change and adjusted p-value are shown.
(XLSX)

**S10 Table. Differential expression of transcripts with GOterm peptidase inhibitor activity.** List of differentially expressed transcripts associated with the Gene Ontology (GO) term Peptidase inhibitor activity (GO:0030414) from *Trichuris muris* eggs embryonated for six and eight weeks. Gene ID, mean expression, log$_2$ fold-change, adjusted p-value and MEROPS family and clan are shown.
(XLSX)

## Acknowledgments

We would like to thank Mr Darran Clements at the Cambridge Stem Cell Institute Imaging Facility for his advice and support on live imaging experiments.

## Author Contributions

**Conceptualization:** Tapoka T. Mkandawire, Richard K. Grencis, Matthew Berriman, María A. Duque-Correa.

**Data curation:** David Goulding, Charlotte Tolley, Stephen R. Doyle, Paul M. Airs, María A. Duque-Correa.

**Formal analysis:** David Goulding, Stephen R. Doyle, Paul M. Airs, María A. Duque-Correa.

**Funding acquisition:** Stephen R. Doyle, Matthew Berriman, María A. Duque-Correa.

**Investigation:** David Goulding, Charlotte Tolley, Tapoka T. Mkandawire, Stephen R. Doyle, Emily Hart, María A. Duque-Correa.

**Methodology:** David Goulding, Charlotte Tolley, Tapoka T. Mkandawire, Stephen R. Doyle, Richard K. Grencis, Matthew Berriman, María A. Duque-Correa.

**Project administration:** Matthew Berriman, María A. Duque-Correa.

**Resources:** Matthew Berriman, María A. Duque-Correa.

**Software:** Stephen R. Doyle.

**Supervision:** Richard K. Grencis, Matthew Berriman, María A. Duque-Correa.

**Validation:** David Goulding, Charlotte Tolley, María A. Duque-Correa.

**Visualization:** David Goulding, Stephen R. Doyle, María A. Duque-Correa.

**Writing – original draft:** David Goulding, Stephen R. Doyle, María A. Duque-Correa.

**Writing – review & editing:** David Goulding, Charlotte Tolley, Tapoka T. Mkandawire, Stephen R. Doyle, Paul M. Airs, Richard K. Grencis, Matthew Berriman, María A. Duque-Correa.

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
