## [Decision Letter · Decision Letter 0]

13 Sep 2024

Dear Dr Duque-Correa,

Thank you very much for submitting your manuscript "Hatching of whipworm eggs induced by bacterial contact is serine-protease dependent" for consideration at PLOS Pathogens. As with all papers reviewed by the journal, your manuscript was reviewed by members of the editorial board and by several independent reviewers. In light of the reviews (below this email), we would like to invite the resubmission of a significantly-revised version that takes into account the reviewers' comments.

All of the reviewers recognized the contribution that this work will have on the field. However, several major concerns were brought up. Two of the reviewers suggested that the conclusions drawn from the microscopy be rewritten and tempered. Especially in light of the recent findings presented by Bond & Huffman 2023 and Robertson, et al., 2023. I would urge the authors to carefully read these references and then review their images. Reviewer 3, expressed willingness to meet with the authors to better clarify his concerns regarding the interpretation of the images. I'm checking with the section editor about whether that is appropriate or not and will get back to you if that is approved. Reviewer 1 raises an important concern regarding the source of the necessary proteases and the data seemingly support bacterial proteases but also nematode proteases. That should be clarified.

We cannot make any decision about publication until we have seen the revised manuscript and your response to the reviewers' comments. Your revised manuscript is also likely to be sent to reviewers for further evaluation.

Sincerely,

Adler R. Dillman, Ph.D.

Academic Editor

PLOS Pathogens

James Collins III

Section Editor

PLOS Pathogens

Michael Malim

Editor-in-Chief

PLOS Pathogens

orcid.org/0000-0002-7699-2064

All of the reviewers recognized the contribution that this work will have on the field. However, several major concerns were brought up. Two of the reviewers suggested that the conclusions drawn from the microscopy be rewritten and tempered. Especially in light of the recent findings presented by Bond & Huffman 2023 and Robertson, et al., 2023. I would urge the authors to carefully read these references and then review their images. Reviewer 3, expressed willingness to meet with the authors to better clarify his concerns regarding the interpretation of the images. I'm checking with the section editor about whether that is appropriate or not and will get back to you if that is approved. Reviewer 1 raises an important concern regarding the source of the necessary proteases and the data seemingly support bacterial proteases but also nematode proteases. That should be clarified.

Reviewer's Responses to Questions

**Part I - Summary**

Reviewer #1: Dear Editors,

Thank you for the opportunity to review the manuscript titled "Hatching of whipworm eggs induced by bacterial contact is serine-protease dependent." The study presents significant contributions to the mechanisms of whipworm egg hatching. This research is particularly relevant given the widespread impact of Trichuris spp. on both human and animal health, especially in neglected areas with limited research funding and increased difficulty in publishing results.

The authors have employed various methodologies, including scanning and transmission electron microscopy, to elucidate the intricate processes underlying whipworm egg hatching. I appreciate the thorough approach taken in this study and look forward to providing constructive feedback that may help enhance the manuscript, particularly regarding the results of the ultrastructure.

Reviewer #2: The manuscript by Goulding et al. is a concise and well-written description of a series of experiments undertaken to better understand the requirement for the successful hatching of eggs of the murine whipworm Trichuris muris. This is an important area, as whipworms are a widespread parasite of humans and animals that cause much disease. A better understanding of this crucial aspect of the parasite biology may lead to novel treatment options. Over the past 10-15 years, a series of papers has demonstrated a requirement for bacterial contact at the polar plugs to initiate hatching, and more recently a requirement for worm-derived chitinase has been proposed. However, there are still substantial gaps in our knowledge, and this paper addresses them through a solid combination of electron microscopy, transcriptomics and functional assays. A requirement for serine proteases is convincingly demonstrated, although whether these are from the bacteria or the worm is equivocal (see also below).

This paper makes an important contribution to the parasitology literature and is suitable for publication in PLOS Pathogens.

Reviewer #3: The strength of the manuscript is in the serine-protease dependence findings.

The weaknesses in the study have to do with the attribution of bacterial causes to technique artifacts induced by application of drying procedures during preparation for electron microscopy. The artifacts under consideration are amplified in the nematode eggshell in ways that have been known for decades but are not widely recognized by authors analyzing TEM & SEM photos.

I am not qualified to comment on other aspects of the MS.

**Part II – Major Issues: Key Experiments Required for Acceptance**

Reviewer #1: (No Response)

Reviewer #2: N/A

Reviewer #3: I do not necessarily recommend additional experiments, but do insist on a complete revision of all references to anatomical aspects of nematode eggshells. There are many clumsy errors in interpretation of eggshell effects that must be corrected.

Rather than recommend experiments, I would be willing to meet with the authors over Zoom, and help them to understand the errors of interpretation. However, I would also require that they completely re-read in detail two of the papers they cited; namely Bond & Huffman 2023 and Robertson, et al., 2023. If they had read these papers carefully before writing the egg anatomy and effects prose, most of the mistakes would not have been made. This is very important, and if the authors are not willing to rewrite most of their anatomical interpretations of bacterial effects on Trichuris eggs, I would recommend rejection.

**Part III – Minor Issues: Editorial and Data Presentation Modifications**

Reviewer #1: Comments:

Introduction:

Lines 69-70: I suggest that the authors include important references that recovered eggs of T. trichiura in non-human primates, as this provides crucial evidence that it can be a zoonotic disease in some countries. As a zoonosis, this impacts the control of egg environmental contamination (refer to line 83).

Additionally, I recommend including this general idea in the discussion and conclusion. This possibility could highlight the sharing of bacterial species in the human and non-human primate microbiota, presenting a fascinating situation that ties into the evolutionary aspects shared by humans and non-human primates.

References: https://doi.org/10.1016/j.vprsr.2017.11.004; https://doi.org/10.3389/fvets.2020.626120

Methodology:

Lines 177, 200, 211, 213, 216: Please specify the exact room temperature in the text.

Line 216 and 225: Change "osmium fixing" to "post-fixing."

Line 219: Change "100 ethanol" to "absolute ethanol."

Results:

Figures 1 and 2: The EM images strongly reinforce the results regarding bacterial interaction. However, I suggest that the authors include the number of samples analyzed/processed, the number of eggs studied, and a quantification (with statistical tests) of the bacteria present on the polar plugs and other parts of the eggshell. This information will significantly enhance the robustness and sustainability of your results.

Figure 2A: Please verify the blue line color of the arrow.

Figure 3: The results are intriguing; however, the authors should be more cautious in the description of figure 3B. The contour of the bacteria outside the halo in the polar plug region, having less electron density, strongly suggests that a material was extracted during ultramicrotomy, which is a common artifact in this process. Additionally, the bacteria present at the base of the plug (indicated near PO in the figure) may not necessarily be present deeper within the plug; this is a two-dimensional image that might create an illusion of a three-dimensional structure where the bacteria could be positioned more laterally of the plug. In summary, I recommend that the authors carefully interpret these results, particularly when describing: "... the PO with sinking and floccular disruption along the full depth of the plug..." and "Degradation proceeds along the full depth of the plug, both with visible sinking and floccular appearance of the PO..."

Discussion:

Lines 600-606: It is challenging to compare results obtained by other authors using fluorescence microscopy with GFP-marked bacteria in hydrated conditions to your SEM results obtained in dried samples. Various factors beyond the biological inference could have influenced the adhesion of bacteria to the polar plugs, especially after all the steps in the conventional scanning electron microscopy protocol (including chemical fixation, post-fixation, washing steps, dehydration, and critical point drying). I recommend either removing this comparison or discussing the possible influences of these parameters on the results.

Thank you for considering these suggestions. I believe they will help enhance the clarity and impact of this important study.

Reviewer #2: I have a few comments for the authors to consider:

1) Figure 1 – how many weeks of embryonation did the eggs undergo before being used for EM imaging of bacterial attachment? Did the authors check whether the stage of embryonation effects bacterial adhesion?

2) The authors show clearly that serine proteases are important for hatching, as treatment of eggs with either inhibitor cocktails or specific serine protease inhibitors (Pefabloc) ablates hatching, whereas inhibition of other protease inhbitors (e.g. E64) have no effect. However, the manuscript lacks clarity regarding the source of these serine proteases. On page 20, the authors propose that, because inhibitor treatment is reversible, it must be acting on bacterial proteases outside the egg and thus bacteria-derived enzymes are key for initiating egg hatching. However, further analyses with RNAseq on Trichuris L1 suggest that parasite proteases are upregulated concomitant to hatching ‘readiness’, and thus likely play a role as well. However, the importance of these parasite-derived molecules is difficult to ascertain. If they played a key role in the hatching process, then at least a degree of hatching would presumably be observed in the absence of bacteria. It could be speculated that the upregulation of these genes in the parasite plays no role in hatching per se, and is instead an adaptation of the parasite to be ‘primed’ to deal with the external environment once hatching is complete.

I appreciate it would be difficult to tease apart these effects – is it possible, for instance, to microinject serine proteases inhibitors into the egg to affect the parasite enzymes whilst leaving the bacterial activity intact on the egg surface?

3) Finally – a general comment – whilst I don’t doubt the importance of gut bacteria in the infectious process of whipworms, it is likely to be complex. There are older papers showing that infection seems to proceed normally in antibiotic treated mice infected with T. muris - Schopf et al. 2002, J. immuol, or pigs infected with T. suis – Mansfield and Urban 1996, Vet Immunol. Immunopathol. These papers are not often cited in this context.

Reviewer #3: I have attached the copy of the PDF I reviewed, which contains several minor points of grammar and word choice and some references to anatomical errors.

PLOS authors have the option to publish the peer review history of their article (what does this mean?). If published, this will include your full peer review and any attached files.

Reviewer #1: **Yes: **Eduardo J. Lopes Torres

Reviewer #2: No

Reviewer #3: No
---

## [Editor Report · Decision Letter 1]

29 Dec 2024

Dear Dr Duque-Correa,

We are pleased to inform you that your manuscript 'Hatching of whipworm eggs induced by bacterial contact is serine-protease dependent' has been provisionally accepted for publication in PLOS Pathogens.

Best regards,

Adler R. Dillman, Ph.D.

Academic Editor

PLOS Pathogens

James Collins III

Section Editor

PLOS Pathogens

Sumita Bhaduri-McIntosh

Editor-in-Chief

PLOS Pathogens

orcid.org/0000-0003-2946-9497

Michael Malim

Editor-in-Chief

PLOS Pathogens

orcid.org/0000-0002-7699-2064

Thank you for your detailed responses to the reviews, and your efforts in revising the manuscript accordingly.
---

## [Editor Report · Acceptance letter]

22 Jan 2025

Dear Dr Duque-Correa,

We are delighted to inform you that your manuscript, "Hatching of whipworm eggs induced by bacterial contact is serine-protease dependent," has been formally accepted for publication in PLOS Pathogens.

Best regards,

Sumita Bhaduri-McIntosh

Editor-in-Chief

PLOS Pathogens

orcid.org/0000-0003-2946-9497

Michael Malim

Editor-in-Chief

PLOS Pathogens

orcid.org/0000-0002-7699-2064